# Constructing and Evaluating a Mitophagy-Related Gene Prognostic Model: Implications for Immune Landscape and Tumor Biology in Lung Adenocarcinoma

**DOI:** 10.3390/biom14020228

**Published:** 2024-02-16

**Authors:** Jin Wang, Kaifan Liu, Jiawen Li, Hailong Zhang, Xian Gong, Xiangrong Song, Meidan Wei, Yaoyu Hu, Jianxiang Li

**Affiliations:** School of Public Health, Suzhou Medical College of Soochow University, Suzhou 215123, China; jinwang93@suda.edu.cn (J.W.); 20214247040@stu.suda.edu.cn (K.L.); 20215247013@stu.suda.edu.cn (J.L.); 20224247032@stu.suda.edu.cn (H.Z.); 20224247016@stu.suda.edu.cn (X.G.); 20234247003@stu.suda.edu.cn (X.S.); 20234247035@stu.suda.edu.cn (M.W.); 20235247033@stu.suda.edu.cn (Y.H.)

**Keywords:** mitophagy, LASSO, prognostic model, lung cancer, immune infiltration

## Abstract

Mitophagy, a conserved cellular mechanism, is crucial for cellular homeostasis through the selective clearance of impaired mitochondria. Its emerging role in cancer development has sparked interest, particularly in lung adenocarcinoma (LUAD). Our study aimed to construct a risk model based on mitophagy-related genes (MRGs) to predict survival outcomes, immune response, and chemotherapy sensitivity in LUAD patients. We mined the GeneCards database to identify MRGs and applied LASSO/Cox regression to formulate a prognostic model. Validation was performed using two independent Gene Expression Omnibus (GEO) cohorts. Patients were divided into high- and low-risk categories according to the median risk score. The high-risk group demonstrated significantly reduced survival. Multivariate Cox analysis confirmed the risk score as an independent predictor of prognosis, and a corresponding nomogram was developed to facilitate clinical assessments. Intriguingly, the risk score correlated with immune infiltration levels, oncogenic expression profiles, and sensitivity to anticancer agents. Enrichment analyses linked the risk score with key oncological pathways and biological processes. Within the model, MTERF3 emerged as a critical regulator of lung cancer progression. Functional studies indicated that the MTERF3 knockdown suppressed the lung cancer cell proliferation and migration, enhanced mitophagy, and increased the mitochondrial superoxide production. Our novel prognostic model, grounded in MRGs, promises to refine therapeutic strategies and prognostication in lung cancer management.

## 1. Introduction

Lung cancer is the leading cause of cancer-related mortality globally and accounts for approximately 18% of all cancer deaths [1]. Non-small cell lung cancer (NSCLC) is the most common lung cancer subtype and comprises two major histological types: lung squamous cell carcinoma (LUSC) and lung adenocarcinoma (LUAD). In addition to conventional therapies such as surgery, chemotherapy, and radiotherapy, targeted therapy and immunotherapy for lung cancer have also developed rapidly in recent years. However, these therapies benefit only a subset of patients and have significant limitations, including side effects and high costs [2,3]. Nearly 70% of patients with NSCLC are initially diagnosed at a locally advanced stage and suffer from a poor prognosis [4]. The 5-year survival rate is less than 3% for patients with advanced NSCLC [5]. Therefore, exploring new diagnostic and prognostic markers is an important way to improve the early diagnosis and prognosis of lung cancer.

Accumulating evidence suggests that cancer is associated with mitochondrial dynamics [6,7]. Mitochondria play important roles in physiological processes ranging from cell metabolism, proliferation, and differentiation to cell survival and apoptosis [8]. Mitochondria are in dynamic equilibrium within cells, constantly changing their morphology to maintain a normal shape, structure, number, and function [9]. Mitophagy is a selective autophagic process essential for cellular homeostasis that eliminates damaged or dysfunctional mitochondria in response to various metabolic stresses, such as hypoxia, growth factor depletion, viral infection, and nutrient deprivation [10,11]. Studies have shown that lethally dysregulated mitophagy plays a role in suppressing tumorigenesis and may provide an avenue for treating various cancers [12].

In this study, we obtained mitophagy-related genes (MRGs) from the GeneCards database. Moreover, we established a prognostic model of MRGs based on the lung adenocarcinoma (LUAD) dataset, in the TCGA database and the other two LUAD datasets in the GEO database, using the least absolute contact and selection operator (LASSO) and Cox regression analysis. This study introduces a novel prognostic model based on MRGs, an approach not previously applied in this context, with implications for personalizing LUAD therapy.

## 2. Materials and Methods

### 2.1. Data Collection

The LUAD data of 572 patients, including 59 normal tissue samples adjacent to cancer tissues, 513 tumor samples, and corresponding clinical information, were retrieved from the Cancer Genome Map (TCGA) database. The expression profile and clinical results are open and accessible. To validate the prognostic model based on the TCGA-LUAD dataset, two LUAD datasets (GSE31210 and GSE13213) were retrieved from the Gene Expression Omnibus (GEO) database as validation datasets. The GSE31210 [13] and GSE13213 [14] datasets contain gene expression and prognostic information from 226 and 117 primary lung adenocarcinoma samples, respectively.

Mitophagy-related genes were extracted from the GeneCards database (https://www.genecards.org/, accessed on 11 January 2023). Briefly, mitophagy-related genes (MRGs) were queried in the GeneCards database with the keyword “mitophagy”, and genes with relevance scores > 1 were further screened to construct a prognostic model for MRGs.

### 2.2. Prognostic Model Construction and Validation

The chi-square test was used to analyze the differences between the training set, the internal test set, and the total dataset in terms of gender, age, tumor stage, depth of invasion (T), lymph node metastasis (N), distal metastasis (M) and smoking history. The univariate Cox model was used to evaluate the associations between continuous expression levels of MRGs and OS. The risk ratio (HR) and *p*-value from univariate Cox regression analysis were used to identify candidate survival-related MRGs. MRGs with an HR > 1 were considered risky, and those with an HR < 1 were defined as protective. The MRGs that met the criterion of a *p*-value < 0.05 were identified as survival-related MRGs and further included in the LASSO and multivariate Cox regression analyses to construct a prognostic model. The risk score for each LUAD patient was calculated based on the expression of MRGs (*Exp_i_*) and Cox coefficients (*coef_i_*): $Risk\;score\;=\sum_{i=1}^{n}{Exp}_{i}\times {coef}_{i}$. All patients in each dataset were divided into high or low-risk groups based on the median risk score. Kaplan–Meier plots were generated to evaluate the survival of patients in each dataset between the high- and low-risk groups. Moreover, multivariate Cox regression analysis was performed to estimate whether the risk score was independent of clinicopathological features. To investigate the performance of the prognostic model in predicting LUAD patient outcomes, the area under the curve (AUC) of the receiver operating characteristic (ROC) was calculated. In addition, the expression of each MRG in the model and its correlation with clinicopathological features were also analyzed.

All analyses were performed with R software (Version 4.1.1, R Foundation for Statistical Computing, Vienna, Austria) and the corresponding fundamental package. The “care” package was used to randomly divide the patients into two datasets at a ratio of 6:4 according to their survival status; these datasets were used as training sets and internal test sets. The “glmnet” package was used for LASSO regression model analysis. In addition, the “survival” and “survminer” packages were used to perform univariate and multivariate Cox analyses and to generate Kaplan–Meier plots. The “TimeROC” package was used to generate the time-dependent receiver-operating characteristic (ROC) curve, and the “survivalROC” package was used to calculate the AUC. Nomogram plots were generated with the “rms” package. 

### 2.3. Enrichment Analysis 

Based on the correlation analysis between the risk score and all mRNAs, gene set enrichment analysis (GSEA) was further performed by using the “ClusterProfiler” package of R software (Version 4.1.1). 

In addition, the differentially expressed genes (DEGs) between the low-risk and high-risk groups were identified with the R 4.1.1 package “limma” with the thresholds of log(fold change) > 1 and a *p*-value < 0.05. The DEGs were further used as input for the DAVID online tool (https://david.ncifcrf.gov/, accessed on 23 June 2023) for pathway and biological process enrichment.

### 2.4. Correlation Analysis 

To further explore the biological role and clinical significance of the MRG prognostic model, correlation analysis was performed between the risk score and scores in each of the following: tumor suppressor gene (TSG) expression, tumor mutation burden (TMB), immune regulatory gene expression, immune cell infiltration, and tumor immune dysfunction and exclusion (TIDE). 

The TSGs were extracted from the TSGene database (https://bioinfo.uth.edu/TSGene/, accessed on 15 July 2023) [15]. The oncogenes were extracted from the ONGene database (http://www.ongene.bioinfo-minzhao.org, accessed on 15 July 2023) [16]. The 11 immune checkpoint genes (ICGs) [17] and 73 immunomodulatory genes (IMGs) [18] were extracted from previous studies. Immune cell infiltration score was obtained from the TIMER2.0 database (http://timer.cistrome.org/, accessed on 18 July 2023) [19]. Moreover, the TIDE score, dysfunction score, and exclusion score of each patient in the datasets were predicted using the TIDE online tool (http://tide.dfci.harvard.edu/, accessed on 18 July 2023) following standard procedures [20]. Correlation analysis was performed with the Spearman method based on the “psych” package.

### 2.5. Antitumor Drug Sensitivity Analysis

The Genomics of Drug Sensitivity in Cancer database (GDSC) was developed by the Sanger Research Institute to collect data on the sensitivity and response of tumor cells to drugs [21]. The “oncoPredict” package was used to calculate the drug sensitivity of each sample in the training and validation datasets based on the GDSC V2 database [22]. 

### 2.6. shRNA Plasmid Construction

*MTERF3* shRNA sequences were designed according to BLOCK-iT™ RNAi Designer (https://rnaidesigner.thermofisher.com/rnaiexpress, accessed on 20 September 2023), and the annealed double-stranded shRNA-encoding oligonucleotides were cloned and inserted into the pGreen vector. After testing the knockdown efficiency of several candidate shRNAs, 2 shRNAs targeting *MTERF3* were selected for subsequent experiments. A scrambled nonspecific control shRNA (shNC) was also cloned, inserted into the same vector, and used as a negative control. 

### 2.7. Cell Culture and Transfection

The human lung cancer cell lines A549 and H1299 were purchased from the American Type Culture Collection (ATCC). All cells were cultured in DMEM (Thermo Fisher Scientific, Waltham, MA, USA) supplemented with 10% FBS (Thermo Fischer Scientific, Inc.) at 37 °C in the presence of 5% CO_2_.

Lung cancer cells were seeded in 6-well plates and grown for 24 h. Then, the cells were transfected with 2.5 μg of shRNA using Lipofectamine 6000 reagent (Beyotime, Shanghai, China) following the manufacturer’s protocol. 

### 2.8. EdU Cell Proliferation Assay

The cells were then incubated with 10 μM EdU for 2 h. The cells were subsequently stabilized with 4% paraformaldehyde and permeabilized using 0.3% Triton X-100 in a PBS environment. A subsequent step involved incubating the cells with a click reaction solution, a product provided by the Beyotime Institute of Biotechnology in China. Within a 24-h timeframe, images of the cells were procured using an inverted fluorescence microscope, and the resulting data were analyzed with the assistance of ImageJ software (Version 1.8.0, National Institutes of Health, Bethesda, MD, USA).

### 2.9. Transwell Migration Assay

Cells from each group were methodically placed in the upper chambers of the Transwell membrane (Corning, New York, NY, USA). Next, 1 mL of medium without FBS and 2 mL of complete medium were added to the bottom chamber. After a 24-h incubation period at 37 °C in an environment with 5% CO_2_, the cells were stabilized in methanol and stained with 0.5% crystal violet for 30 min. The final stage involved washing the cells three times in the upper chamber with phosphate-buffered saline (PBS, provided by Gibco, Grand Island, NY, USA). The cells were then imaged using a microscope and evaluated with the use of ImageJ software (version 1.8.0).

### 2.10. Mitophagy Fluorescence Assay

After 48 h of transfection with shRNAs, an LC3-GFP lentivirus and Mito-RFP lentivirus (HanBio, Shanghai, China) were cotransfected into lung cancer cells for 48 h. The cells were then imaged using a microscope and evaluated with the use of ImageJ software (version 1.8.0).

### 2.11. MitoSOX Assay

The cells were incubated with 5 μM MitoSOX Green probe (Invitrogen, Eugene, OR, USA) for 30 min. After the cells were washed twice with PBS, the mean fluorescence intensity (MFI) was measured via flow cytometry (FACS Canto II, BD Biosciences, San José, CA, USA). The mitochondrial superoxide level was proportional to the MitoSOX MFI.

### 2.12. Statistical Analysis

Statistical analyses were conducted using GraphPad software (Version 8.3.0, GraphPad Software, San Diego, CA, USA), and the data are represented as the mean ± standard deviations. To ascertain the existence of statistically significant differences between the means of two or more groups, Student’s *t*-test and analysis of variance (ANOVA) were employed. After performing one-way ANOVA, the Tukey method was applied for post hoc multiple comparisons. All the statistical tests were two-tailed, and a *p*-value of less than 0.05 was considered to indicate statistical significance.

## 3. Results

### 3.1. Data Collection

Clinical data from three LUAD cohorts were obtained from the TCGA and GEO databases. The demographic and clinical data for the training, internal testing, and independent validation sets are summarized in Table 1. After filtering out the samples with missing clinical information from the TCGA-LUAD dataset, a total of 504 LUAD patients, including 183 living and 321 dead patients at the end of follow-up, were included in this study (median follow-up: 2.474 years). The dataset was randomly divided into a training set (*n* = 303) and an internal testing set (n = 201). As expected, no significant differences were found in the major clinicopathological features between the training, testing, and entire TCGA-LUAD datasets (Table 1). In addition, this study included two GEO datasets comprising 226 and 117 LUAD patients, namely, GSE31210 and GSE13213, respectively, which included 37.81% and 15.49% of the deaths at the end of follow-up (median follow-up time was 4.720 years and 5.306 years respectively).

### 3.2. Construction and Validation of the Prognostic Model According to MRGs in LUAD Patients

Based on the GeneCards dataset, a total of 272 mitophagy-related genes (MRGs) were screened with the criteria of a relevance score > 1 (Appendix A). Moreover, the differential expression analysis demonstrated that 110 of these MRGs, including 77 upregulated and 33 downregulated genes, were differentially expressed in LUAD samples compared with normal adjacent tissues with the criteria of |log2 (fold change)| > 1 and *p*-value < 0.05 (Appendix A). Forty prognosis-related MRGs were identified based on the TCGA training set using univariate Cox regression analysis (Figure 1A). Consequently, LASSO-penalized Cox analysis further identified 20 MRGs for multivariate analysis (Figure 1B,C). The multivariate Cox proportional hazard model was built stepwise using the likelihood-ratio forward method to reach the highest significance. Hence, 12 MRGs were further screened to construct a risk model to assess the prognostic risk of patients with LUAD: risk score = (0.467 × OSBPL5 Exp) + (−0.329 × VPS13D Exp) + (0.255 × LMAN1 Exp) + (−0.846 × ATG4A Exp) + (0.410 × HSPA9 Exp) + (−0.212 × STEAP3 Exp) + (0.408 × CHMP2A Exp) + (−0.260 × OGT Exp) + (0.463 × UBC Exp) + (0.416 × MTERF3 Exp) + (−0.580 × PRKCD Exp) + (0.360 × PLSCR1 Exp) (Figure 1D). TimeROC curves demonstrated that the risk score was a significant predictor of the OS of LUAD patients, with the AUCs greater than 0.730 at 1–5 years (Figure 1E). The samples in the training set were classified into low- and high-risk groups based on the median risk score. The distributions of risk scores between the low-risk and high-risk groups and the survival status and survival time of patients in the two different risk groups are depicted in Figure 1F. KM survival analysis indicated that the low-risk group has significantly favorable overall survival (OS) (Figure 1G). The relative expression of the 12 MRGs for each patient is shown in Figure 1H. Importantly, when three other survival indicators, namely DSS (disease-specific survival), DFI (disease-free interval), and PFI (progression-free interval), were considered, the KM survival analysis indicated that the low-risk group had a significantly favorable outcome for the LUAD patients (Appendix A).

To further verify the accuracy and reliability of the prognostic model obtained from the training set, we applied it to the testing set and the other two independent validation cohorts, viz. GSE31210 and GSE13213. By using the same prognostic model, the classifier could also successfully subdivide patients in the internal testing set (n = 201) into high-risk or low-risk groups with remarkable differences in OS (*p* = 0.008; Appendix A). In addition, the same observation was also found in the entire TCGA-LUAD dataset (Figure 2A) and in the GSE31210 and GSE31213 validation cohorts (Figure 2B,C). Additionally, timeROC curves indicated that the risk score was an effective predictor for the OS of LUAD patients in the entire TCGA-LUAD, GSE31210, and GSE31213 datasets, with almost all AUCs greater than 0.700 at 1–5 years (Figure 2D–F). Consistent with the results demonstrated in the training set, the KM survival analysis indicated that the mitophagy risk score was a significant risk factor for the OS of LUAD patients in the above three datasets (all *p* < 0.010, Figure 2G–I).

### 3.3. Mitophagy Risk Score Is an Independent and Practical Indicator for OS

As depicted in Appendix A, the mitophagy risk score was related to several clinicopathological features, including lymph node metastasis, invasion depth, stage, and smoking history, in the entire TCGA-LUAD dataset. Moreover, the differences in expression between the 12 MRGs in the model and different clinicopathological features were also analyzed, and the results indicated that several MRGs were related to different features (Appendix A–E). To assess whether the mitophagy risk score is an independent indicator in LUAD patients, the effect of each clinicopathologic feature on OS was analyzed by univariate Cox regression (Figure 3A). As shown in Figure 3B, after multivariate adjustment, the risk score remained a powerful and independent factor in the entire TCGA-LUAD dataset. Moreover, the risk score was verified to be an independent factor based on the GSE31210 (Appendix A) and GSE13213 (Appendix A) datasets. The discrepancies in OS stratified by the clinicopathologic features were analyzed between the low- and high-risk groups in the entire TCGA-LUAD dataset. According to the results for the invasion depth subgroup (Figure 3C,D) and lymph node metastasis subgroup (Figure 3E,F) the OS of the low-risk score group was superior to that of the high-risk group.

To ensure the robustness and practicability of the 12-MRG prognostic model, prognostic nomograms that incorporate significant clinicopathological characteristics and the risk score derived from our model were established based on the TCGA-LUAD (Figure 4A), GSE31210 (Figure 4C) and GSE13213 (Appendix A) datasets. Each variable can be located on the respective axis, and a line can be drawn upward to determine the number of points awarded for each variable. The sum of these points is located on the ‘Total Points’ axis, from which a line can be drawn downward to the survival axes to determine the likelihood of 1-year, 3-year, or 5-year OS. To assess the predictive performance of the nomograms, we calculated the bootstrap C-index and created calibration plots. The C-indexes for our nomogram were 0.749, 0.779, and 0.745 for the TCGA-LUAD, GSE21310, and GSE13213 datasets, respectively (Appendix A). The calibration plot demonstrated a strong agreement between the predicted and observed survival probabilities (Appendix A), which indicated that the nomogram was well-calibrated. Furthermore, ROC curve analysis was conducted to assess the specificity and sensitivity of the nomogram’s predictive performance (AUC ≥ 0.760 at 1, 3, and 5 years) in the TCGA-LUAD (Figure 4B) and GSE31210 (Figure 4D) datasets.

### 3.4. Mitophagy Risk Score Is Associated with the Immune Landscape

Based on the XCELL algorithm, the mitophagy risk score was shown to be associated with the infiltration of multiple immune and stromal cells (Figure 5A), including Th1/2 CD4+ T cells and mast cells (r = −0.411 in TCGA-LUAD, Figure 5B), as well as common lymphoid progenitors and common myeloid progenitors (r = −0.287 in TCGA-LUAD, Figure 5E). Additionally, the risk score was associated with the immune score (r = −0.321 in TCGA-LUAD, Figure 5C) and the microenvironment score (r = −0.317 in TCGA-LUAD, Figure 5D). In addition, based on the TIDE online tool, risk scores were significantly positively correlated with the TIDE score, T cell exclusion score (r = −0.321 in TCGA-LUAD, Figure 5H), as well as the infiltration of cancer-associated fibroblasts and myeloid-derived suppressor cells (MDSC, r = −0.321 in TCGA-LUAD, Figure 5I) in three datasets (Figure 5F). However, risk scores were significantly negatively correlated with the T cell dysfunction score (r = −0.321 in TCGA-LUAD, Figure 5G) and the infiltration of tumor-associated macrophages. Overall, these results imply that the mitophagy risk score may be associated with the poorer therapeutic efficacy of immune checkpoint inhibitors.

### 3.5. The Mitophagy Risk Score Is Associated with Drug Sensitivity and Oncogene Expression

To further explore the relationship between the risk score and antitumor drug sensitivity, we first analyzed the relationship between the risk score and several commonly mutated genes as drug targets. The results indicated that LUAD patients with mutations in the KRAS, RET, or TP53 genes had higher risk scores (Figure 6A). Using the OncoPredict package and GDSC data as the training set, the sensitivity of TCGA-LUAD samples to 198 antitumor drugs was calculated. Further correlation analysis revealed that the risk score significantly related to the sensitivity to multiple drugs (Figure 6B), especially doramapimod with r = 0.453 in the TCGA-LUAD dataset (Figure 6C), 0.266 in the GSE13213 dataset (Figure 6D) and 0.450 in the GSE31210 dataset (Figure 6E). Due to doramapimod being a p38-MAPK inhibitor, we further analyzed the correlation between the risk score and the expression of genes in the p38-MAPK pathway in the three datasets. The results revealed that risk score was significantly related to multiple genes in the p38-MAPK pathway in the three datasets (Figure 6F and Appendix A). In addition, further correlation analysis revealed that the risk score was positively correlated with the expression of many oncogenes (Figure 6J), mainly CENPW, CDK1, CDC6, ECT2, CCNB1, FOXM1, BIRC5 and several other oncogenes, in the TCGA-LUAD (Figure 6G), GSE13213 (Figure 6H) and GSE31210 (Figure 6I). Conversely, the risk score was negatively correlated with the expression of several TSGs in the three datasets (Appendix A). Moreover, the risk score was significantly correlated with TMB (r = 0.251 in TCGA-LUAD, Appendix A).

### 3.6. MRG Risk Score Is Associated with Cancer Progression

GSEA analysis was performed to investigate the biological processes and pathways potentially related to the MRG risk score. As depicted in Figure 7A,B, the MRG risk score was related to multiple cancer-related biological processes in the three datasets, including DNA replication (NES = 2.896 in TCGA-LUAD, Figure 7B), mitochondrial gene expression (NES = 2.759 in TCGA-LUAD, Figure 7B), double-strand break repair (NES = 2.584 in TCGA-LUAD, Figure 7C) and cellular respiration (NES = 2.553 in TCGA-LUAD, Figure 7C). Additionally, the risk score was related to several important pathways (Figure 7D), mainly including spliceosome (NES = 2.951 in TCGA-LUAD, Figure 7E), cell cycle (NES = 2.789 in TCGA-LUAD, Figure 7E), DNA replication (NES = 2.697 in TCGA-LUAD, Figure 7F) and mismatch repair (NES = 2.236 in TCGA-LUAD, Figure 7F). Moreover, differential expression analysis identifies DEGs between the high- and low-risk groups, and the further enrichment analysis revealed that these DEGs were significantly enriched in several important biological processes (Figure 7G) and pathways (Figure 7H), including gene expression by genetic imprinting, cell adhesion, negative regulation of apoptotic process, cellular response to hypoxia and drug metabolism, as well as notch and hippo signaling pathways.

### 3.7. MTERF3 Contributes to Cancer Progression 

Among the MRGs identified in the constructed risk model, MTERF3 was strongly correlated with the MRG risk score in 3 datasets (Figure 8A and Appendix A). Survival analysis revealed that patients with lower MTERF3 expression in the TCGA-LUAD dataset had longer overall survival (Appendix A). When considering disease-specific survival and disease-free interval, a better prognosis was found for patients with low MTERF3 expression (Appendix A). MTERF3 expression was greater in tumors than in normal tissues in multiple LUAD datasets (Figure 8B). The GSEA demonstrated that MTERF3 is related to many cancer-related KEGG pathways (Figure 8C) and biological processes (Figure 8D), including cell adhesion molecules (CAMs, NES = −2.649), DNA replication (NES = 2.422), cell cycle (NES = 2.518), proteasome (NES = 2.610), DNA replication (NES = 2.429, mitochondrial translation (NES = 2.423), macrophage activation (NES = −2.202), extracellular matrix assembly (NES = −2.546), as well as several other vital terms. Further correlation analysis reveals that MTERF3 expression is significantly correlated with multiple oncogenes (Figure 8E). Additionally, MTERF3 expression was positively correlated with tumor stemness (Figure 8G) and tumor mutation burden (TMB, Figure 8H), as well as the imputed sensitivity of doramapimod (Figure 8I) in the TCGA-LUAD dataset. 

### 3.8. MTERF3 Regulates Proliferation, Migration, and Mitophagy in LUAD Cell

To evaluate the biological function of MTERF3 in LUAD cells, we constructed shRNA plasmids to knock down this gene (Appendix A). The EdU assay revealed that MTERF3 knockdown attenuated the proliferation of A549 and H1299 cells (Figure 9A,B). Considering cell migration, the transwell migration assay indicated that the knockdown of MTERF3 significantly reduced the number of migrated cells (Figure 9C,D). Per its effects on mitophagy, the fluorescence labeling assay showed significant colocalization of the autophagy marker LC3 and mitochondria in lung cancer cells after MTERF3 knockdown (Figure 9E), and the green/red (LC3/mitochondria) ratio was significantly increased in MTERF3 knockdown cells (Figure 9F,G). Further analysis revealed that MTERF3 knockdown significantly amplified the level of mitoSOX, a mitochondrial superoxide indicator, in lung cancer cells (Figure 9H,I).

## 4. Discussion

Since Otto Warburg discovered “aerobic glycolysis” in solid tumor cells more than 80 years ago [23], mitochondria have been shown to play an increasingly important role in cancer biology. Mitophagy has been reported to be a key mechanism of metabolic reprogramming and aerobic glycolysis regulation within tumor cells and holds significant promise in the treatment of various cancer types [24]. Abnormalities in mitophagy result in the survival of damaged and mutated mitochondria, which promotes the emergence and progression of malignancy. Our study focused on the link between lung cancer and mitophagy to create a prognostic model for mitophagy-related genes based on LASSO/Cox regression and further confirmed the reliability and clinical significance of this model.

Recently, several studies have constructed risk prediction models for different cancers, including pancreatic cancer [25], glioblastoma [26], hepatocellular cancer [27], and lung cancer [28], based on MRGs. The prognostic model constructed based on these 12 MRGs had better predictive accuracy and robustness than the other models, and patients at high risk had worse prognoses in both the training and validation cohorts. The 1–4 year survival AUC values for the training set were greater than 0.770 and greater than 0.700 in the entire TCGA-LUAD dataset as well as in two independent GEO datasets. This model has excellent accuracy compared with other mitophagy-related prognostic models reported previously (AUC = 0.695) [28]. After univariate and multivariate Cox regression analyses were performed, the risk score was identified as an independent prognostic factor. Importantly, we constructed a nomogram based on multivariate analysis, and the score calculated based on this nomogram was able to predict patient prognosis well and was suitable for use in clinical practice. By integrating this model into clinical workflows, it can significantly refine the precision of lung cancer prognostication, guiding clinicians in devising personalized treatment plans that align closely with the individual patient risk profiles, thereby optimizing clinical outcomes and potentially improving overall survival rates.

Functionally, based on the differential expression gene analysis and enrichment analysis between the high- and low-risk groups, we found that the risk score was associated with multiple cancer-related signaling pathways and biological processes. Moreover, the risk score was negatively correlated with the expression of many tumor suppressor genes and positively correlated with the expression of multiple oncogenes, indicating its oncogenic role in LUAD. The tumor microenvironment represents the noncancerous cells and other components present in a tumor and the continuous interaction between tumor cells. It plays a vital role in tumor initiation, progression, metastasis, and response to therapy [29]. Autophagy can negatively or positively regulate the immune evasion of cancer cells through the degradation of immune checkpoint proteins and antigens, release of cytokines, and generation of antigens [30]. Similarly, mitophagy plays a crucial role in modulating anticancer immune responses [31,32,33]. Further analysis suggested that the risk score was negatively correlated with the infiltration of multiple immune cells and the TIDE score, indicating that the risk score regulates the immune microenvironment and immune response. The results revealed a significant negative correlation between the risk score and the infiltration of many types of immune and stromal cells, including mast cells (MCs), common myeloid progenitors, hematopoietic stem cells, T cells, and B cells. MCs localize at the margins of tumors and the TME, commonly around vessels [34]. Within the TME, mast cells (MCs) can either inhibit or promote tumor growth. Upon activation, they generate proinflammatory responses and attract innate and acquired immune cells, driving antitumor reactions. Alternatively, their presence can promote tumor progression by secreting VEGF, aiding angiogenesis, and MMP9, degrading the ECM, thus facilitating metastasis [35]. Mast cells accumulate in the tumor stroma of multiple cancer types, and their increase has been reported to be associated with patient prognosis and regulate multiple biological processes [36,37]. 

Among the 12 MRGs in the model, many, such as UBC [38] and ATG4A [39], have been reported to play significant regulatory roles in autophagy. Maintaining proteostasis, as well as the integrity and functionality of organelles, is critical to maintaining cellular homeostasis and vitality. Autophagy is a key homeostatic pathway that facilitates the degradation and recycling of cellular constituents [40,41]. The MTERF3 protein, which is the most highly conserved member of the human MTERF protein family, consists of 417 amino acid residues and five conserved mTERF motifs [42]. MTERFs are encoded by nuclear genes and are transported from the nucleus to the cytoplasm, where they localize to mitochondria [43]. These proteins participate in regulating mitochondrial gene replication, transcription, and translation by binding to mitochondria [44]. Previous studies have shown that mammalian MTERF3 acts as a negative regulator of mitochondrial DNA (mtDNA) transcription [45]. MTERF3 is essential for mammalian embryonic development, and knockout of this protein results in delayed embryonic development and embryonic lethality [45]. Abnormal mtDNA transcription caused by the inactivation of MTERF3 in the myocardium and skeletal muscle tissue can lead to severe respiratory chain defects and reduced oxidative phosphorylation [46]. Additionally, MTERF3 was reported to promote cell growth and irradiation resistance by regulating interleukin (IL)-6 and IL-11 in colorectal cancer cells [47]. Here, MTERF3 was identified as a vital MRG in our risk prediction model, and it is upregulated in lung cancer and may serve as an oncogene. Further functional assays revealed that MTERF3 deficiency significantly attenuates the proliferation and migration ability of lung cancer cells while promoting mitophagy and the generation of mitochondrial superoxide. Although the current literature highlights the potential role of the regulation of mitochondria in cancer therapy, there is a lack of reports on the mechanism of mitochondria as a drug target or on the upstream regulatory mechanisms involved.

This study has several limitations. The data used in this study were collected mainly from publicly available databases, and additional prospective clinical data are needed to demonstrate the utility of our prognostic risk model. In addition, further in vitro and in vivo experiments are required to explore the specific roles and regulatory mechanisms of MRGs in a LUAD model, which may provide new opportunities for the treatment of lung cancer.

## Figures and Tables

**Figure 1 biomolecules-14-00228-f001:**
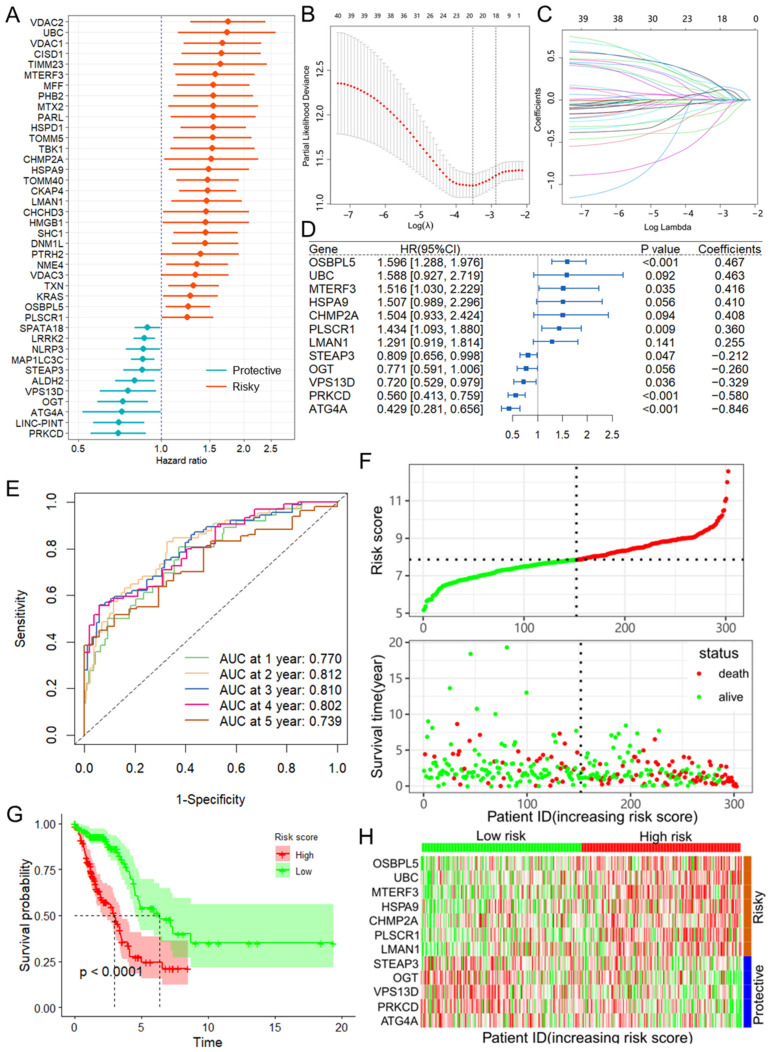
Construction of the prognostic model of MRGs. (**A**) Univariate Cox regression analysis for the selection of MRGs correlated with the OS of HCC patients. (**B**,**C**) LASSO Cox regression analysis identified a total of 20 MRGs for overall survival. (**D**) The forest plot shows the multivariate Cox regression analysis of 12 MRGs. MRGs: mitophagy-related genes. (**E**) TimeROC curves for 1-, 2-, 3-, 4-, and 5-year OS in the training set. (**F**) Risk score distribution and survival status of the training group. (**G**) Kaplan–Meier curve of OS in the training group. (**H**) Heatmap showing the expression of 12 MRGs in the training sets. MRGs: mitophagy-related genes; TimeROC: time-dependent receiver operating curve; OS: overall survival.

**Figure 2 biomolecules-14-00228-f002:**
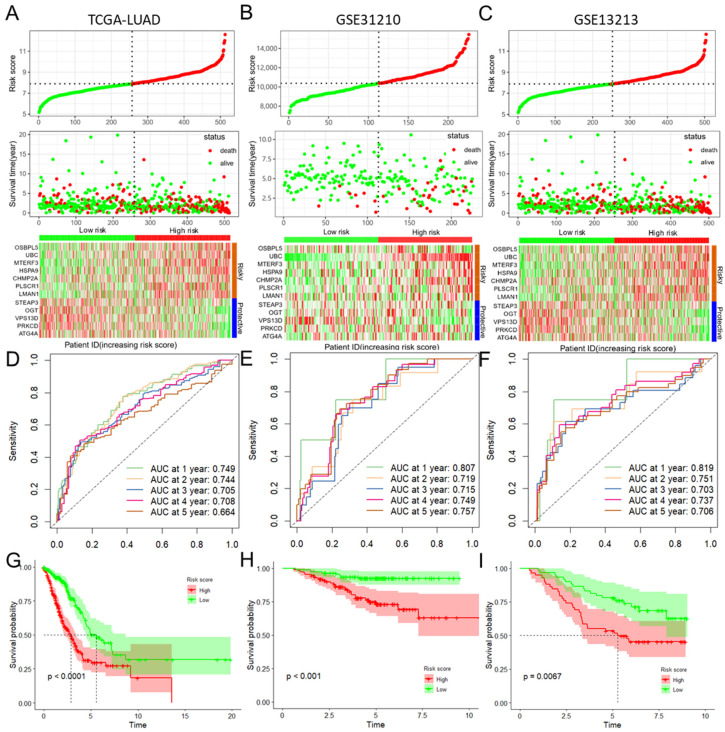
Validation of the prognostic model with 12 MRGs constructed from the training dataset. Risk score distribution, survival status, and expression of 12 MRGs in the entire TCGA-LUAD dataset (**A**) and two external validation datasets, viz. GSE31210 (**B**) and GSE31213 (**C**) datasets. TimeROC curves for 1-, 2-, 3-, 4-, and 5-year OS in the entire TCGA-LUAD dataset (**D**), as well as in the GSE31210 (**E**) and GSE31213 (**F**) datasets. KM curve of OS in the entire TCGA-LUAD dataset (**G**), as well as GSE31210 (**H**) and GSE31213 (**I**) datasets. MRGs: mitophagy-related genes; TimeROC: time-dependent receiver operating curve; LUAD: lung adenocarcinoma; TCGA: The Cancer Genome Atlas; OS: overall survival.

**Figure 3 biomolecules-14-00228-f003:**
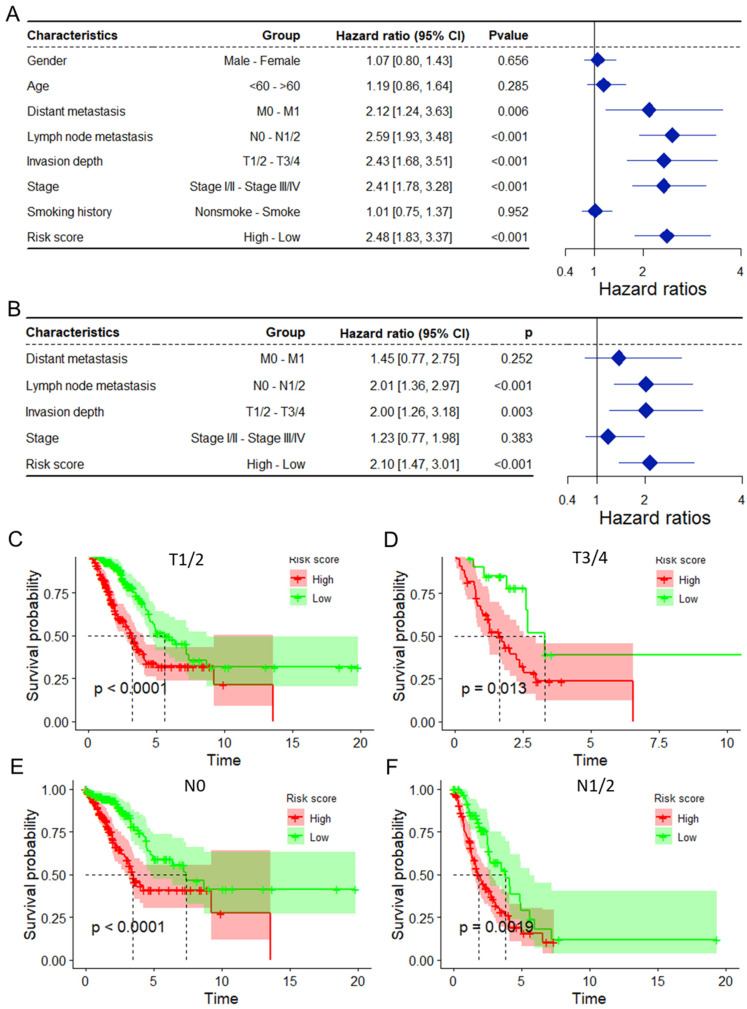
The mitophagy risk score was found to be an independent prognostic factor for OS in the entire TCGA-LUAD dataset. Univariate (**A**) and multivariate (**B**) Cox regression analyses of the risk score and clinicopathological features for overall survival in the entire TCGA-LUAD dataset. (**C**,**D**) KM analysis of OS stratified by invasion depth. (**E**,**F**) KM analysis of OS stratified by lymph node metastasis. TCGA: The Cancer Genome Atlas; LUAD: lung adenocarcinoma; OS: overall survival.

**Figure 4 biomolecules-14-00228-f004:**
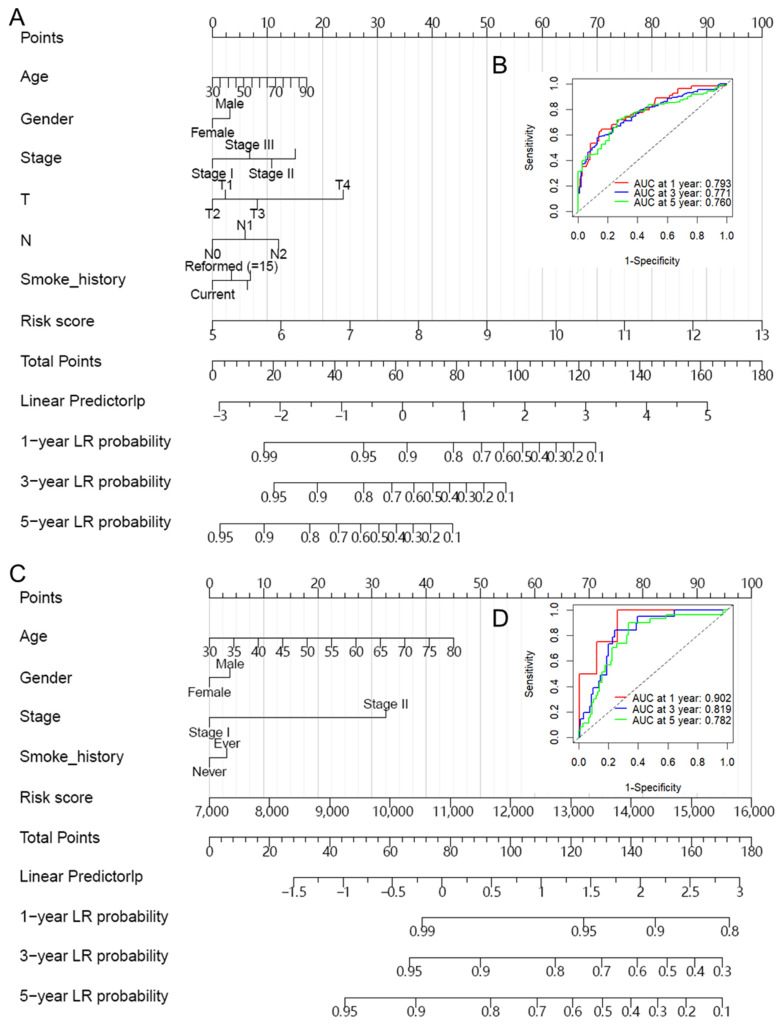
Nomogram for predicting 1-year, 3-year, and 5-year overall survival in lung adenocarcinoma patients. (**A**) The nomogram consists of the 12-gene risk score and 6 clinical indicators based on the entire TCGA-LUAD dataset. (**B**) ROC curve analysis was conducted to assess the specificity and sensitivity of the nomogram’s predictive performance in the TCGA-LUAD dataset. (**C**) The nomogram consists of the MRG risk score and 4 clinical indicators based on the GSE13210 dataset. (**D**) ROC curve analysis was conducted to assess the specificity and sensitivity of the nomogram’s predictive performance in the GSE13210 dataset. The points from these variables are combined to determine the location of the total points. The total points projected on the bottom scales indicate the probabilities of 1-year, 3-year, and 5-year overall survival. ROC: receiver operating characteristic; TCGA: The Cancer Genome Atlas; LUAD: lung adenocarcinoma.

**Figure 5 biomolecules-14-00228-f005:**
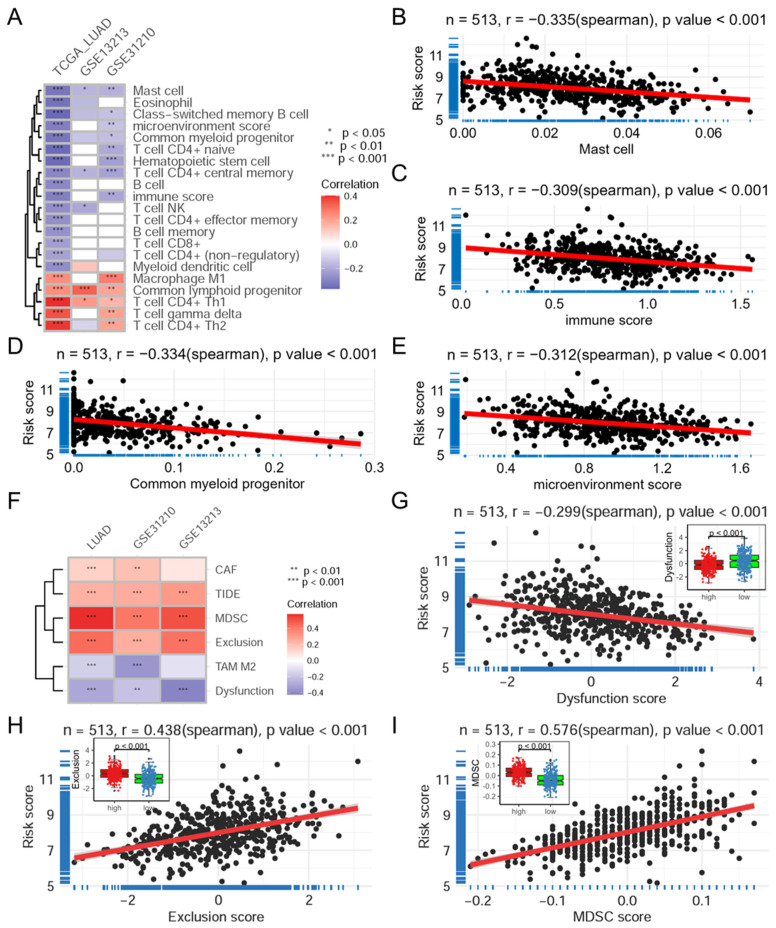
The mitophagy risk score is associated with the immune landscape. (**A**) Heatmap showing the correlation between the 12-MRG risk score and immune cell infiltration based on the XCELL algorithm in 3 datasets. Scatter plots show the correlation between risk score and mast cell infiltration (**B**), immune score (**C**), microenvironment score (**D**) and common myeloid progenitor infiltration (**E**). (**F**) Heatmap showing the correlation analysis between risk scores and scores based on the TIDE online tool. Scatter plots showing the correlations between the risk score and dysfunction score (**G**), exclusion score (**H**) and infiltration of myeloid-derived suppressor cells (**I**). The red line represents the linear fitting. TCGA: The Cancer Genome Atlas; LUAD: lung adenocarcinoma; TIDE: tumor immune dysfunction and exclusion.

**Figure 6 biomolecules-14-00228-f006:**
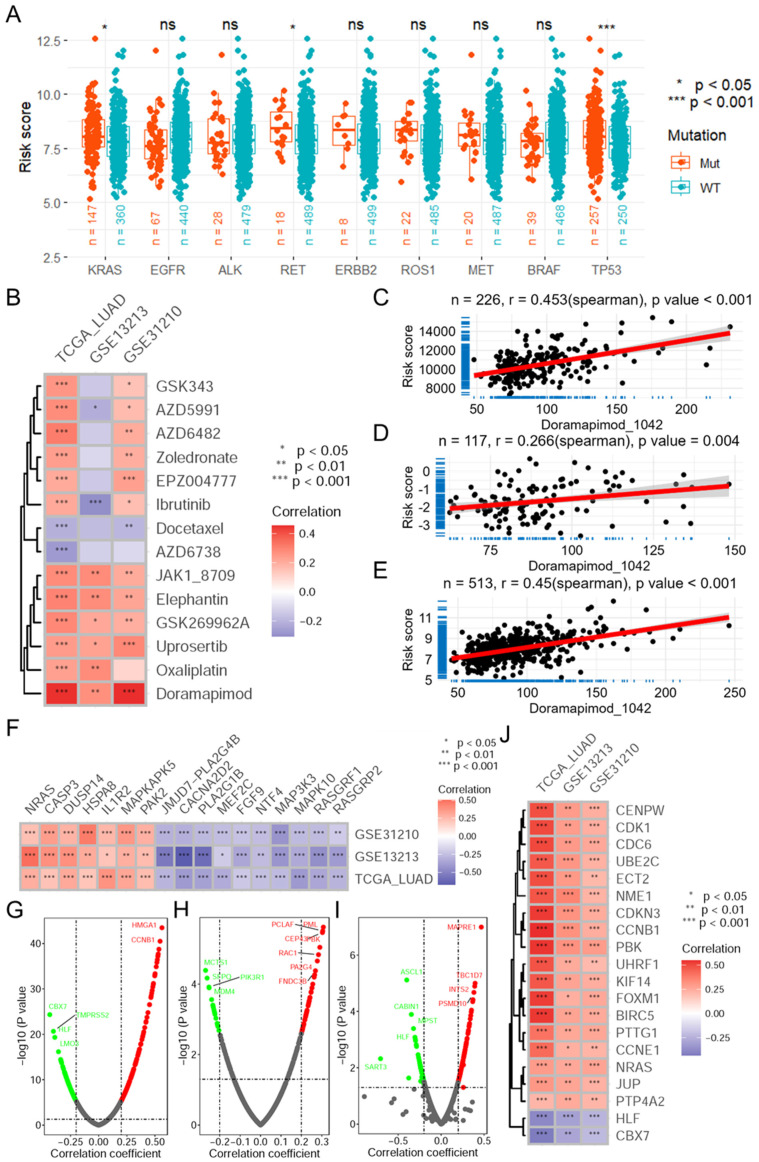
The mitophagy risk score correlates with the sensitivity to antitumor drugs and the expression of oncogenes. (**A**) Risk score distribution of mutated genes commonly used as drug targets in lung cancer. (**B**) Heatmap showing the correlation between the risk score and the imputed sensitivity to antitumor drugs based on the GDSC 2.0 database and the “OncoPredict” package in 3 datasets. Scatter plots showing the correlation between the risk score and the sensitivity to doramapimod in the TCGA-LUAD (**C**), GSE13213 (**D**) and GSE31210 (**E**) datasets. (**F**) Heatmap showing the correlation between the risk score and the expression of genes in the p38-MAPK pathway. Volcano plots showing the correlation between risk score and the expression of oncogenes in the TCGA-LUAD (**G**), GSE13213 (**H**) and GSE31210 (**I**) datasets. (**J**) Heatmap showing the correlation between the risk score and the expression of oncogenes in 3 datasets. TCGA: The Cancer Genome Atlas; LUAD: lung adenocarcinoma; MRGs: mitophagy-related genes.

**Figure 7 biomolecules-14-00228-f007:**
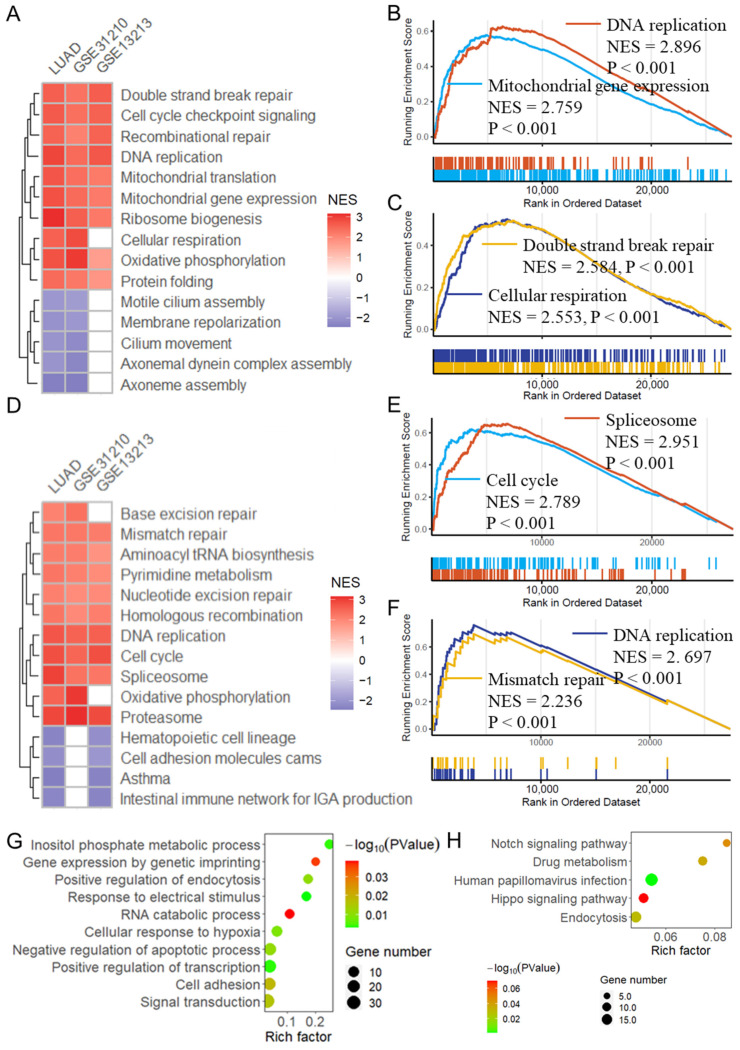
Biological processes and KEGG pathways related to the mitophagy risk score. (**A**) Heatmap showing the GSEA results for the biological process of the 12-MRG risk score in the 3 datasets. (**B**,**C**) GSEA plots showing that the risk score is related to DNA replication, mitochondrial gene expression, double-strand break repair, and cellular respiration. (**D**) Heatmap showing the GSEA results for the KEGG pathways associated with the MRG risk score in the 3 datasets. (**E**,**F**) GSEA plots showing that the risk score is related to spliceosome, cell cycle, DNA replication, and mismatch repair. Bubble plots showing that the differentially expressed genes between high and low risk groups enriched in several important biological processes (**G**) and KEGG pathways (**H**). LUAD: lung adenocarcinoma; GSEA: gene set enrichment analysis; MRG: mitophagy-related gene.

**Figure 8 biomolecules-14-00228-f008:**
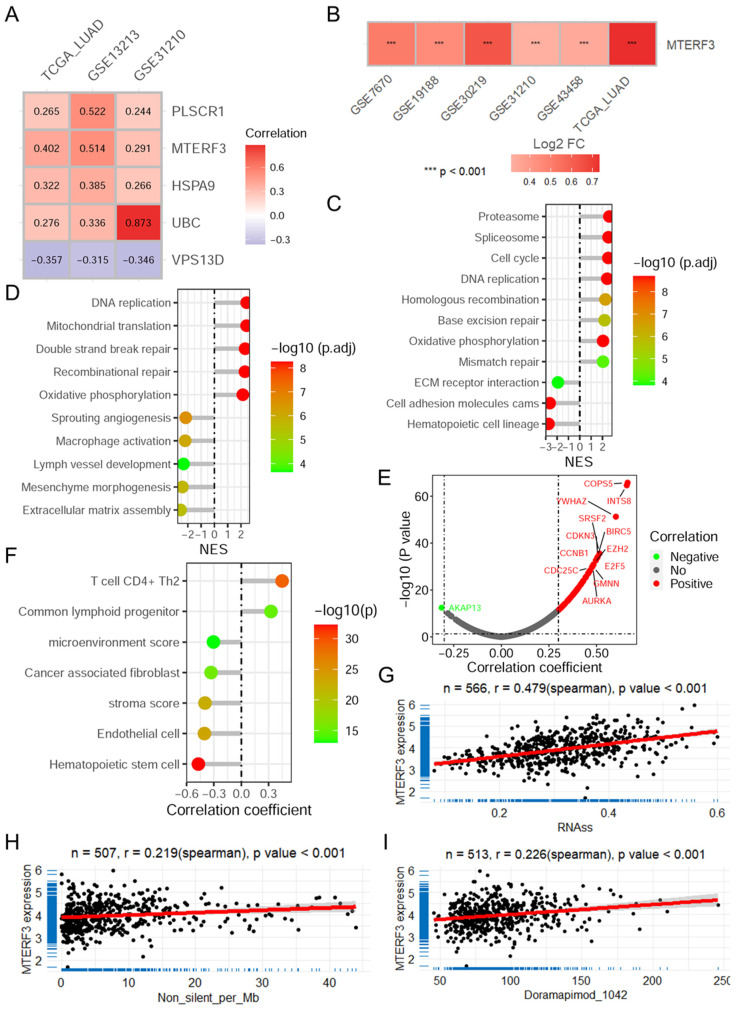
MTERF3 is highly expressed in LUAD and related to cancer progression. (**A**) Heatmap showing the correlation between the risk score and MRGs in the risk model in the 3 datasets. (**B**) Heatmap showing the change in MTERF3 expression in tumors compared with normal tissue in multiple datasets. Lollipop plots showing the results of GSEA results of MTERF3 for KEGG pathway (**C**) and biological process (**D**) in the TCGA-LUAD dataset. (**E**) The volcano plot shows the results of the correlation analysis between the expression of MTERF3 and oncogenes. (**F**) The lollipop plot shows the correlation between MTERF3 and immune cell infiltration. Scatter plots showing the correlations between MTERF3 expression and tumor stemness (**G**), TMB (**H**) and the imputed sensitivity of doramapimod (**I**) in the TCGA-LUAD dataset. The red line represents the linear fitting. TCGA: The Cancer Genome Atlas; LUAD: lung adenocarcinoma; TMB: tumor mutation burden; GSEA: gene set enrichment analysis; MRGs: mitophagy-related genes.

**Figure 9 biomolecules-14-00228-f009:**
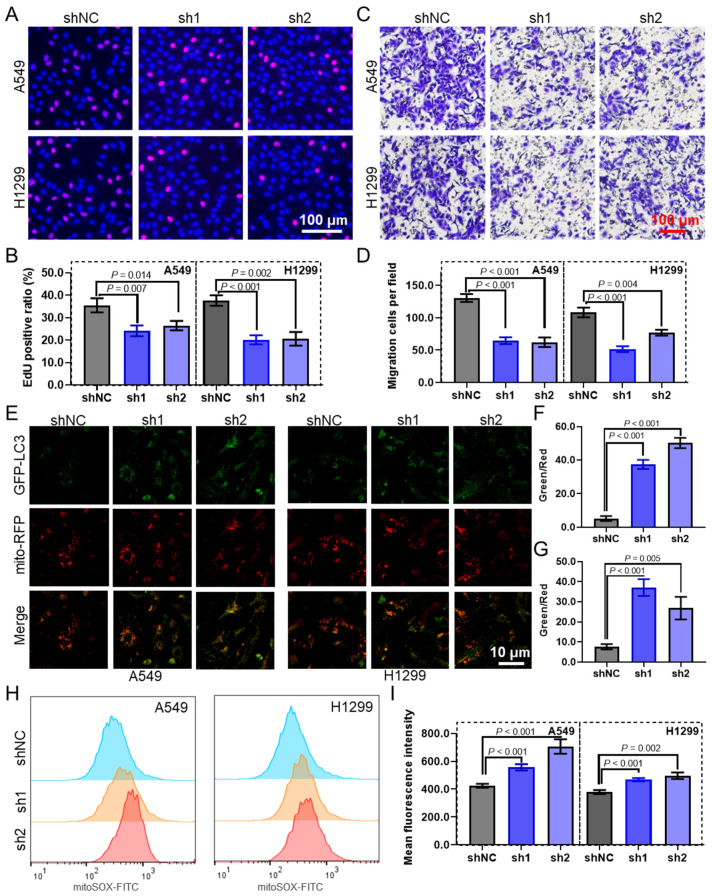
MTERF3 promotes cell proliferation, migration, and disulfidptosis in LUAD. Representative images (**A**) and the quantified results (**B**) of the EdU cell proliferation assay in LUAD cells with MTERF3 knockdown. Representative images (**C**) and the quantified results (**D**) of the transwell cell migration assay in LUAD cells with MTERF3 knockdown. Representative images (**E**) and the quantified results (**F**,**G**) of the fluorescence labeling assay in LUAD cells with MTERF3 knockdown, green represents LC3, and red represents mitochondria. Representative images (**H**) and the quantified results (**I**) of the mitoSOX assay measured by flow cytometry in LUAD cells with MTERF3 knockdown. LUAD: lung adenocarcinoma.

**Table 1 biomolecules-14-00228-t001:** Clinical features of the LUAD patients in the training, testing, and validation sets.

Characteristics	TCGA-LUAD Cohort				Independent Validation Cohorts	
	Training Set (60%) *n* = 303	Testing Set (40%) *n* = 201	All Data *n* = 504	χ^2^ *p*-Value	GSE31210 *n* = 226	GSE13213 *n* = 117
Age						
<60	203 (68.35%)	133 (67.51%)	336 (68.02%)	0.981	118 (52.21%)	65 (55.56%)
≥60	94 (31.65%)	64 (32.49%)	158 (31.98%)		108 (47.79%)	52 (44.44%)
Gender						
Female	160 (52.81%)	110 (54.73%)	270 (53.57%)	0.914	121 (53.54%)	57 (48.72%)
Male	143 (47.19%)	91 (45.27%)	234 (46.43%)		105 (46.46%)	60 (51.28%)
Stage						
Stage I	163 (54.88%)	107 (53.77%)	270 (54.44%)	>0.999	168 (74.34%)	79 (67.52%)
Stage II	72 (24.24%)	48 (24.12%)	120 (24.19%)		58 (25.66%)	13 (11.11%)
Stage III	47 (15.82%)	33 (16.58%)	80 (16.13%)			25 (21.37%)
Stage IV	15 (5.05%)	11 (5.53%)	26 (5.24%)			
T						
T1	109 (36.09%)	60 (30.15%)	169 (33.73%)	0.214		54 (46.15%)
T2	164 (54.30%)	105 (52.76%)	269 (53.69%)			50 (42.74%)
T3	23 (7.62%)	22 (11.06%)	45 (8.98%)			8 (6.84%)
T4	6 (1.99%)	12 (6.03%)	18 (3.59%)			5 (4.27%)
M						
M0	199 (92.99%)	136 (93.15%)	335 (93.06%)	0.998		
M1	15 (7.01%)	10 (6.85%)	25 (6.94%)			
N						
N0	192 (65.08%)	132 (67.01%)	324 (65.85%)	0.998		87 (74.36%)
N1	57 (19.32%)	38 (19.29%)	95 (19.31%)			8 (6.84%)
N2	45 (15.25%)	26 (13.20%)	71 (14.43%)			22 (18.80%)
N3	1 (0.34%)	1 (0.51%)	2 (0.41%)			
Smoke history						
Never	37 (12.76%)	35 (17.86%)	72 (14.81%)		115 (50.88%)	
Current	68 (23.45%)	51 (26.02%)	119 (24.49%)	0.743	111 (49.12%)	
Reformed (≤15)	80 (27.59%)	48 (24.49%)	128 (26.34%)			
Reformed (>15)	105 (36.21%)	62 (31.63%)	167 (34.36%)			
Status						
Live	196 (64.69%)	125 (62.19%)	321 (63.69%)	0.850	125 (62.19%)	191 (84.51%)
Dead	107 (35.31%)	76 (37.81%)	183 (36.31%)		76 (37.81%)	35 (15.49%)

## Data Availability

All data generated and described in this article are available from the corresponding web servers and are freely available to any scientist wishing to use them for noncommercial purposes without breaching participant confidentiality. All codes and R-packages used in the study are publicly available and have been disclosed in Methods or are available from the corresponding authors on reasonable request.

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
