# Peer review of "Constructing and Evaluating a Mitophagy-Related Gene Prognostic Model: Implications for Immune Landscape and Tumor Biology in Lung Adenocarcinoma"

_biomolecules, 2024, doi:10.3390/biom14020228_

Round 1
Reviewer 1 Report
Comments and Suggestions for Authors
I like the approach of this study and also the combination of bioinformatic analyses and cell culture - based functional studies of a candidate gene. Some minor comments:
- statistics: Can the authors comment whether and how p values were corrected for multiple testing?
-concept: The genes used for the model are not exclusively associated with highly selective mitophagy but also with the more general autophagy system (i.e. the umbrella pathway). This is e.g. shown by appearance of ATG4A, UBC etc. Thus, the authors' findings could also reflect a contribution of other facets of autophagy (as compared to sole mitophagy) for prognosis, therapy and disease manifestation in general. Maybe mitochondrial degradation is only one aspect of a broader role of proteostasis in lung cancer? The authors should discuss this idea and cite 1-2 papers on this topic (autophagy in cancer).
- experimental approach: Which criteria were used to pick MTERF3 as candidate gene for functional studies? Did it display the highest correlation? Or which other arguments were used in favor of this gene? Please include these criteria.
- Figure 9A,C, E: please include size bar in one picture per panel
- discussion: Could MTERF3 represent an attractive drug target? Are there drugs or compounds know to affect its transcriptional activity? Please discuss.
- LASSO or lasso?
- l. 256: Is the mitophagy risk score in deed related to smoking history? Fig 3A seems to show that its not. Please clarify!
- Shortly mention or discuss which role mitophagy (or also autophagy) may play in the observed immune landscape alterations.
l. 319 better ...'and oncogene expression'
l. 366: better: 'cell adhesion, negative regulation of...'
l. 410 better : '..plasmids to knock down this gene...'
l. 454: better 'many tumor suppressor genes'
Comments on the Quality of English Languageonly some very minor suggestions:
- l.49: lethal mitophagy; better ' lethally dysregulated mitophagy'? (since mitophagy is not deadly per se.)
l.83: low-risk groups with the median; better 'based on the median'
l.105: DEGs were further input into; better 'were further used as input for'
l. 128 'the annealed shRNA double stranded was cloned....', better 'the annealed double stranded shRNA-encoding oligonucleotides were cloned...'
l.147 the resultant data was, better 'the resulting data were'
l.197 replace 'forty' by '40'
Author Response
Dear professor,
We would like to express our sincere gratitude for your insightful comments and suggestions. Following your advice, we have thoroughly revised the manuscript to address the concerns raised. We will respond to your questions point by point:
- statistics: Can the authors comment whether and how p values were corrected for multiple testing?
Reply: Thanks for your suggestion. As requested, we have addressed the issue of multiple comparisons in our study. In the revised manuscript, we have expanded the “Statistical analysis” section to clearly describe the methods we used to correct for multiple testing. Specifically, after performing one-way ANOVA using GraphPad, we applied the Tukey method for post hoc multiple comparisons.
-concept: The genes used for the model are not exclusively associated with highly selective mitophagy but also with the more general autophagy system (i.e. the umbrella pathway). This is e.g. shown by appearance of ATG4A, UBC etc. Thus, the authors' findings could also reflect a contribution of other facets of autophagy (as compared to sole mitophagy) for prognosis, therapy and disease manifestation in general. Maybe mitochondrial degradation is only one aspect of a broader role of proteostasis in lung cancer? The authors should discuss this idea and cite 1-2 papers on this topic (autophagy in cancer).
Reply: We greatly appreciate the thoughtful feedback regarding our manuscript. Per your suggestion, we have expanded the discussion to contemplate the broader implications of mitophagy in cancer in the revised manuscript.
“Within the 12 MRGs in the model, many genes have been reported to play significant regulatory roles in autophagy, such as UBC[31] and ATG4A[32]. Maintaining proteostasis as well as the integrity and functionality of organelles is critical to cellular homeostasis and vitality. Autophagy is a key homeostatic pathway that facilitates the degradation and recycling of cellular constituents[33,34].”
- experimental approach: Which criteria were used to pick MTERF3 as candidate gene for functional studies? Did it display the highest correlation? Or which other arguments were used in favor of this gene? Please include these criteria.
Reply: Thank you for raising this point. Indeed, our selection of MTERF3 was based on the correlation analysis between the risk scores and the 12 MRGs in the model across three datasets. We identified this gene by taking the intersection of the top three genes with the highest correlation in all three datasets. In the revised manuscript, we have included a new Figure S10, which displays the top three genes with the highest correlation and the results of the intersection.
- Figure 9A,C, E: please include size bar in one picture per panel
Reply: Thanks for your suggestion, we have addressed these points and made the corresponding modifications in the revised manuscript.
- discussion: Could MTERF3 represent an attractive drug target? Are there drugs or compounds know to affect its transcriptional activity? Please discuss.
Reply: Thank you for your question. Upon review, we found no current literature on drugs or compounds targeting MTERF3's transcriptional activity. The manuscript has been updated to reflect this discussion.
“Although current literature highlights the potential role of its regulation of mitochondria in cancer therapy, there is a lack of reports on its mechanisms as a drug target or on the upstream regulatory mechanisms involved.”
- LASSO or lasso?
Reply: We appreciate your attention to detail. This should be capitalized, we have made modifications to the entire text
- l. 256: Is the mitophagy risk score in deed related to smoking history? Fig 3A seems to show that its not. Please clarify!
Reply: Thank you for your question. Indeed, Figure 3A displays the results from our univariate Cox analysis, which suggests that smoking history is not a prognostic factor in the TCGA-LUAD dataset. The narrative in the text pertains to the results shown in Table S2, which highlight the variations in risk scores across samples with different clinicopathological features.
- Shortly mention or discuss which role mitophagy (or also autophagy) may play in the observed immune landscape alterations.
Reply: Thank you for raising this point. We have supplemented the discussion in our manuscript to address the role of mitophagy, as well as autophagy, in the observed alterations of the immune landscape.
“Autophagy can negatively or positively regulate the immune evasion of cancer cells through the degradation of immune checkpoint proteins and antigens, release of cytokines, and generation of antigens [27]. Similarly, mitophagy plays a crucial role in modulating anticancer immune responses [28-30].”
l.319 better ...'and oncogene expression'
l.366: better: 'cell adhesion, negative regulation of...'
l.410 better : '..plasmids to knock down this gene...'
l.454: better 'many tumor suppressor genes'
Reply:The suggested corrections have been made, thank you.
Comments on the Quality of English Language
- l.49: lethal mitophagy; better ' lethally dysregulated mitophagy'? (since mitophagy is not deadly per se.)
l.83: low-risk groups with the median; better 'based on the median'
l.105: DEGs were further input into; better 'were further used as input for'
l.128 'the annealed shRNA double stranded was cloned....', better 'the annealed double stranded shRNA-encoding oligonucleotides were cloned...'
l.147 the resultant data was, better 'the resulting data were'
l.197 replace 'forty' by '40'
Reply:The suggested corrections have been made, thank you.
Reviewer 2 Report
Comments and Suggestions for Authors In this manuscript, entitled "Constructing and Evaluating a Mitophagy-Related Gene Prog- 2 nostic Model: Implications for Immune Landscape and Tumor 3Biology in Lung Adenocarcinoma" authors using bioinformatic analyses and prognostic models tried to find mitophagy related genes that correlates with the progression of lung cancer. They found that the MTERF3 gene links to cancer progression and with some biological experiments in cell cultures verified their hypothesis.
I found the manuscript to be overall well written and much of it to be well described. The presented data provides very useful information for people who works with mitophagy in carcinogenesis field. However, I have some concerns:
This study has several similarities with a recently published study
Dai D, Liu L, Guo Y, Shui Y, Wei Q. A Comprehensive Analysis of the Effects of Key Mitophagy Genes on the Progression and Prognosis of Lung Adenocarcinoma. Cancers (Basel). 2022 Dec 22;15(1):57. doi: 10.3390/cancers15010057. PMID: 36612054; PMCID: PMC9817891), however in the present study the authors's analyses concluded in a specific gene (MTERF3) that is linked with cancer progression.
In order authors to strongest their conclusion, I suggest to also make the same experiments as in Figure 9 (B, D, E) by knock down the mitophagy related gene VPS13D, that is not correlated with high cancer risk and they can used it as control.
Also, I suggest a better presentation and explanation of Nonograms in order to be easy understood from non bioinformatics people.
Also, authors have to explain further (in Discussion section) based on their analyses their statement for personalizing LUAD therapy.
Comments on the Quality of English LanguageMinor comments:
-Results 3.2 in line 199 delete the "were identified"
-Figure 1 F, What represent the x axon?
-Figure 1 H. Is there wrong labeling in the risk bar? Since in all previous graphs the High risk is denoted with red and low risk with yellow color.
-Figure 3 F. Please write the lymph node metastasis that this graph refer to
-Discussion. Line 485, rewrite the "iderntified"
Author Response
In order authors to strongest their conclusion, I suggest to also make the same experiments as in Figure 9 (B, D, E) by knock down the mitophagy related gene VPS13D, that is not correlated with high cancer risk and they can used it as control.
Reply: We are grateful for your suggestion. The primary aim of our study was to construct and validate a risk prediction model based on mitophagy-related genes (MRGs). Subsequent in vitro experiments were conducted to analyze the biological functions of key genes within the model. In the revised manuscript, we have included the rationale for selecting MTERF3 as a key gene (see Figure S10). We thank you again for your recommendation, and in our future experiments, we will further analyze and discuss the biological functions of these MRGs in lung cancer, including VPS13D.
Also, I suggest a better presentation and explanation of Nonograms in order to be easy understood from non bioinformatics people.
Reply: Thank you for your valuable suggestion. We understand the importance of clear presentation and explanation of our nomogram for it to be accessible to readers who may not have a background in bioinformatics. To this end, we have revised our manuscript to include a more detailed description.
“To ensure the robustness and practicability of the 12-MRGs prognostic model, prognostic nomograms that incorporates significant clinicopathological characteristics and the risk score derived from our model was established based on TCGA LUAD (Figure 4A) and GSE31210 (Figure 4C) datasets, as well as GSE13213 (Figure S6D). Each variable can be located on the respective axis, and a line can be drawn upwards to determine the number of points awarded for each variable. The sum of these points is located on the ‘Total Points’ axis, from which a line can be drawn downwards to the survival axes to determine the likelihood of 1-year, 3-year, and 5-year overall survival. To assess the predictive performance of the nomograms, we calculated the bootstrap C-index and created calibration plots. The C-index for our nomogram was 0.749, 0.779 and 0.745 in TCGA-LUAD, GSE21310 and GSE13213 separately (Figure S6A-C). The calibration plot demonstrates a strong agreement between predicted and observed survival probabilities (Figure S6A-C), which indicates that the nomogram is well-calibrated. Furthermore, ROC curve analysis was conducted to assess the specificity and sensitivity of the nomogram’s predictive performance (AUC ≥ 0.760 at 1, 3 and 5 years) in TCGA LUAD (Figure 4B) and GSE31210 (Figure 4D) datasets.”
Also, authors have to explain further (in Discussion section) based on their analyses their statement for personalizing LUAD therapy.
Reply: Thank you for your constructive comments and for giving us the opportunity to elaborate on the significance of our analyses for personalizing treatment in lung adenocarcinoma (LUAD). In accordance with your suggestion, we have expanded the Discussion section to provide a more comprehensive explanation of how our risk prognosis model can inform and tailor individual treatment strategies for LUAD.
We have elucidated the potential of the model in stratifying patients into different risk categories, which could significantly influence therapeutic decision-making, from the selection of surgical options to the administration of adjuvant therapies. Furthermore, we have discussed how the integration of the nomogram, with its high degree of accuracy, into clinical practice could enable oncologists to predict patient prognosis more effectively and thus personalize treatment approaches.
We believe that these additions will provide readers with a clearer understanding of the practical implications of our research for enhancing the personalization of LUAD therapy.
Comments on the Quality of English Language
-Results 3.2 in line 199 delete the "were identified"
Reply:Corrected, thank you.
-Figure 1 F, What represent the x axon?
Reply:Thank you for your meticulous review. Indeed, Figure 1F and H share an X-axis, and we apologize for any confusion this has caused. We have separated them in the revised manuscript for a clearer presentation of the results.
-Figure 1 H. Is there wrong labeling in the risk bar? Since in all previous graphs the High risk is denoted with red and low risk with yellow color.
Reply:Thank you for your careful review. Indeed, the high and low-risk labels in Figure 1H were incorrectly marked, and we have now rectified this error.
-Figure 3 F. Please write the lymph node metastasis that this graph refer to
Reply:Thank you for your careful review. We have added the label in the revised manuscript。.
-Discussion. Line 485, rewrite the "iderntified"
Reply: We appreciate your attention to detail in identifying the spelling error. It has been corrected in the revised manuscript. Thank you for bringing this to our attention.
Round 2
Reviewer 2 Report
Comments and Suggestions for Authors
I would like to thank the authors that they have answered in all my questions. Therefore, I agree to publish the revised manuscript in Biomolecules.
Comments on the Quality of English Language
Moderate modifications are needed to the English language.
Author Response
Dear professor,
Thanks for your affirmation of our responses in the first round review.
We have entrusted the manuscript to a professor who is a native speaker of English for language editing, and we have also re-examined the revised content for confirmation.
Thanks again for your suggestions on the language of this manuscript. We hope this revised manuscript has addressed your concern.
Yours Sincerely!